# Human Hematopoietic Stem Cell Engrafted IL-15 Transgenic NSG Mice Support Robust NK Cell Responses and Sustained HIV-1 Infection

**DOI:** 10.3390/v15020365

**Published:** 2023-01-27

**Authors:** Shawn A. Abeynaike, Tridu R. Huynh, Abeera Mehmood, Teha Kim, Kayla Frank, Kefei Gao, Cristina Zalfa, Angel Gandarilla, Leonard Shultz, Silke Paust

**Affiliations:** 1Department of Immunology and Microbiology, The Scripps Research Institute, La Jolla, CA 92037, USA; 2Scripps Research Translational Institute, La Jolla, CA 92037, USA; 3Division of Internal Medicine, Scripps Clinic/Scripps Green Hospital, La Jolla, CA 92037, USA; 4The Jackson Laboratory, Bar Harbor, ME 04609, USA

**Keywords:** humanized mice, NSG, NK cells, IL-15, HIV-1, immunotherapy

## Abstract

Mice reconstituted with human immune systems are instrumental in the investigation of HIV-1 pathogenesis and therapeutics. Natural killer (NK) cells have long been recognized as a key mediator of innate anti-HIV responses. However, established humanized mouse models do not support robust human NK cell development from engrafted human hematopoietic stem cells (HSCs). A major obstacle to human NK cell reconstitution is the lack of human interleukin-15 (IL-15) signaling, as murine IL-15 is a poor stimulator of the human IL-15 receptor. Here, we demonstrate that immunodeficient NOD.Cg-*Prkdc^scid^ Il2rg^tm1Wjl^*/SzJ (NSG) mice expressing a transgene encoding human IL-15 (NSG-Tg(IL-15)) have physiological levels of human IL-15 and support long-term engraftment of human NK cells when transplanted with human umbilical-cord-blood-derived HSCs. These Hu-NSG-Tg(IL-15) mice demonstrate robust and long-term reconstitution with human immune cells, but do not develop graft-versus-host disease (GVHD), allowing for long-term studies of human NK cells. Finally, we show that these HSC engrafted mice can sustain HIV-1 infection, resulting in human NK cell responses in HIV-infected mice. We conclude that Hu-NSG-Tg(IL-15) mice are a robust novel model to study NK cell responses to HIV-1.

## 1. Introduction

Currently, more than 37 million people are living with HIV-1 worldwide. An effective vaccine or cure is still not available, and treatment thus remains life-long anti-retroviral therapy. Humanized mice are essential for the investigation of HIV-1 pathogenesis, prevention, and therapy [1,2]. Among humanized mouse models, the bone marrow–liver–thymus (BLT) model, which supports multilineage hematopoiesis, is the only humanized mouse model supporting class I and class II human leukocyte antigen (HLA)-restricted T cell responses and sustained HIV-1 infection [3,4,5], albeit Rag1KO.IL2RγcKO.NOD mice expressing HLA class II-DR4 molecule (DRAG mice) [6] have been used to evaluate the susceptibility of T follicular helper cells [7] to HIV-1 infection and HIV-1 tissue distribution [8]. However, NK cells, which contribute to rapid antibody-dependent and -independent antiviral responses, develop in reduced frequencies in these mice. Therefore, a model that allows for the robust engraftment of human NK cells is needed for NK-cell-focused mechanistic studies and the development of potential novel therapeutics for HIV-1.

Along with T and B cells, NK cells are an essential antiviral weapon of the immune system. They respond quickly to infection, kill infected cells without prior activation, engage multiple targets simultaneously, and secrete large amounts of antiviral cytokines [9,10]. NK cells also mediate crosstalk with the adaptive immune system through antibody-dependent cell-mediated cytotoxicity (ADCC) [11,12,13]. Mounting epidemiological and experimental evidence points to an essential role for NK cell effector functions in host resistance to HIV: (i) NK cells are associated with protection from infection in HIV-1-exposed seronegative individuals, who present with vigorous antiviral NK cell activity [14,15,16,17,18,19,20,21,22,23,24]; (ii) circulating NK cells expand during early acute HIV-1 infection and inhibit HIV-1 replication [25,26,27,28,29]; (iii) specific combinations of NK cell killer immunoglobulin-like receptor (KIR) genes expressed in conjunction with their human leukocyte antigen (HLA) ligands are associated with slower HIV-1 disease progression and lower viral set-point [14,16,22,26,27,29,30,31,32,33,34,35,36,37,38,39,40]; (iv) NK-cell-mediated HIV specific effector functions exert immune selection pressure, resulting in viral escape [27,28,29,31]; and (v) an RV144 secondary immune correlate analysis showed that the low plasma anti-HIV-1 Env IgA antibodies and high levels of ADCC inversely correlate with infection risk [41].

Current humanized mouse models express low levels of human cytokines vital for the survival and proliferation of human NK cells [42]. One of these critical cytokines is IL-15, the lack of which results in the reduced frequency of NK cells in HSC-humanized NSG mice. IL-15 is a pleiotropic cytokine produced predominantly by epithelial cells, fibroblasts, keratinocytes, monocytes, macrophages, and dendritic cells [43,44,45,46,47,48,49,50]. IL-15 can bind directly to the receptor complex composed of the IL-15Rβ and the common γ chain of the IL-2 receptor [51,52]. More often, IL-15 is trans-presented by antigen-presenting cells on the high-affinity receptor IL-15Rα [53,54,55,56] to the βγ receptor complex on NK cells. The binding of IL-15/IL-15Rα to the βγ receptor complex initiates signaling through the recruitment and phosphorylation of Janus kinases (JAKs) 1 and 3, which in turn phosphorylate and activate STAT5 [57]. STAT5 has many target genes, including *Mcl1,* which is required for the survival of NK cells [58]. Importantly, STAT5 promotes the expression of the T-box transcription factors EOMES and, to a lesser extent, T-bet [59], both of which play key roles in the development, maturation, and function of NK cells [60]. In addition, STAT5 is correlated with the expression of several NK cell-derived effector molecules, including IFN-γ, perforin, and granzyme B [59]. It has been demonstrated in mice that the loss of either IL-15 [61] or IL-15Rα [62] leads to a reduction in NK cell frequency. While human IL-15 supports survival and robust effector functions of murine NK cells [63,64,65], murine-derived IL-15 binds with much lower affinity to the human IL-15R [65] and human NK cells do not develop robustly in earlier models of humanized mice [56].

Efforts to augment levels of IL-15 in humanized mice, including the administration of exogenous human IL-15 and human IL-15Rα [66], or hydrodynamic injection of plasmid DNA encoding human IL-15 [67], result in only temporary increases in NK cell proliferation and survival. To address these deficiencies, commercially available human IL-15 transgenic mice [68] have been generated. However, their IL-15 serum levels are 10–20-fold higher than those found in healthy male and female, young and old humans, where the interquartile range for serum IL-15 is 0–9 pg/mL, with 90% of healthy humans below 20 pg/mL [69]. Chronically elevated levels of serum IL-15 (>150 pg/mL) may cause lymphoproliferative disease and limit survival in humanized NOG-IL-15 mice [68,70]. In addition, IL-15 knock-in [71,72] mice have also been reported; however, some express additional cytokines to boost T cell development, and neither are commercially available at this time. Moreover, whether these IL-15 knock-in or transgenic mouse models are susceptible to HIV-1 infection has only been demonstrated in one of these models thus far [73].

Recently, Dr. Leonard Shultz at The Jackson Laboratory developed an NSG [74,75] mouse model expressing a transgenic 200 kbp construct of the complete human IL-15 gene. In this study, we demonstrate that NSG-Tg(IL-15) mice have physiological serum levels of human IL-15. When humanized with umbilical-cord-blood (UCB)-derived HSCs, Hu-NSG-Tg(IL-15) mice support robust and long-term human NK cell development, survival, and proliferation. The human NK cells are phenotypically normal and populate lymphoid and non-lymphoid tissue. Additionally, splenic NK cells from Hu-NSG-Tg(IL-15) mice displayed normal cytotoxic function. Importantly, gross and histological examination of Hu-NSG-Tg(IL-15) revealed no signs of GVHD. Finally, Hu-NSG-Tg(IL-15) mice sustained HIV-1 infection and replication, which was reciprocated with robust antiviral responses from human NK cells.

## 2. Materials and Methods

### 2.1. Mice

NOD.Cg-*Prkdc^scid^Il2rg^tm1wjl^* Tg(IL-15) mice (NSG-Tg(IL-15) (Stock No: 030890)) were developed in Dr. Leonard Shultz’s research lab by pronuclear injection of a ~200 Kbp bacterial artificial chromosome (BAC) designated Chori BACPAC RP11-620F3 containing the human *IL-15* gene. NSG-Tg(IL-15) mice (Stock No: 030890), CD34+ humanized NSG-Tg(IL-15), and NSG mice were provided by or purchased from The Jackson Laboratory. Recipient mice were sublethally irradiated at 4–12 weeks of age and intravenously injected with CD34^+^ enriched human cord-blood-derived cells. The reconstitution efficiency of each animal was independently determined by flow cytometric analysis of PBMCs from whole blood collected from the submandibular vein. PBMCs were isolated by density gradient centrifugation using Ficoll-Paque (GE Healthcare, Chicago, IL, USA) as per the manufacturer’s protocol. Cells were washed and resuspended in 0.2 mL of PBS supplemented with 2% FBS, incubated with murine and human Fc Block (Becton Dickinson (BD), Franklin Lakes, NJ, USA) for 10 min at room temperature (RT), and then stained with antibodies specific to murine CD45 (30-F11; BioLegend, San Diego, CA, USA) and human CD45 (H130; BioLegend, San Diego, CA, USA), before acquisition through flow cytometry.

### 2.2. Human IL-15 ELISA

The Human IL-15 Quantikine ELISA Kit (Cat#D1500 R&D Systems, Minneapolis, MN, USA) was used to quantify levels of human IL-15 in the sera of Hu-NSG-Tg(IL-15) and humanized NSG mice according to the manufacturer’s guidelines, and optical density was determined using the SpectraMax^®^ iD3 microplate reader set to 450 nm.

### 2.3. Flow Cytometry

Blood, spleen, liver, bone marrow (BM), and the gastrointestinal (GI) tract were harvested at experimental endpoints. Single-cell suspensions were generated by mechanical disruption using a 40 μm nylon mesh. Immune cells were then isolated by density gradient centrifugation using Ficoll-Paque (GE Healthcare, Chicago, IL, USA) as per the manufacturer’s protocol. Immune cells were washed once in sterile PBS + 2% FBS, resuspended in PBS + 2% FBS, incubated with murine and human Fc Block (BD) for 10 min at RT, and stained with antibodies listed in Appendix A. For permeabilization, the FoxP3 permeabilization buffer kit (Tonbo Biosciences, San Diego, CA, USA) was used according to the manufacturer’s instructions. The acquisition was performed on the 5 L 16UV-16V-14B-10YG-8R Cytek™ Aurora flow cytometer (Cytek Biosciences, Fremont, CA, USA). Analyses of FCS files were performed using FlowJo (Becton, Dickinson and Company, Franklin Lakes, NJ, USA). Data were graphed and statistical analyses were performed using Graphpad Prism (San Diego, CA, USA).

### 2.4. Hematoxylin and Eosin Staining

Tissue specimens were excised from patients/mice and fixed in 10% Neutral Buffered Formalin (NBF) for 18–24 h. Fixed tissues were dehydrated through an ethanol gradient and subsequently embedded using HistoStar™ Embedding Workstation. Formalin-fixed paraffin-embedded (FFPE) blocks were cut into 5 µm sections using the Thermo Scientific™ HM 325 Rotary Microtome. Slides were deparaffinized in xylene, rehydrated through an ethanol gradient, and washed in deionized water prior to staining with hematoxylin (VWR Cat#95057-844) for 5 min. Slides were then washed with deionized water, dipped in acid alcohol, washed with deionized water, and dipped in 1% lithium carbonate bluing solution. Slides were washed with 70% ethanol, stained with eosin (VWR Cat#95057-848), rehydrated through an ethanol gradient, and soaked in xylene for 15 min before mounting with VectaMount permanent mounting medium (Vector Laboratories Cat# H-5000). Slides were visualized with the Vectra^®^ Polaris™ and analyzed with Phenochart (v1.0.12).

### 2.5. Immunohistochemistry

Multiplex fluorescent IHC was performed using the Manual Opal 4-Color IHC Kit Cat# NEL810001KT) with primary antibodies targeting human NKp46 (R&D Systems, Bio-Techne, Minneapolis, MN, USA, clone 195314, catalog MAB1850), CD3 (Thermo Fisher Scientific, Waltham, MA, USA, clone SP7, catalog RM-9107-S0), and CD45 (Cell Signaling Technology, Danvers, MA, USA, clone D9M8I, catalog 13917S). Slides were baked at 60 °C for 1 h and deparaffinized in xylene. Slides were rehydrated by ethanol gradient and fixed in 10% NBF. Antigen retrieval was conducted by microwave treatment in Antigen Retrieval buffer pH6 (AR6) for NKp46 and CD3 and pH9 (AR9) for CD45. Slides were stained by consecutively incubating with blocking solution, primary antibody, secondary antibody, and opal reagent. Microwave treatment and staining steps were repeated for each primary antibody, followed by DAPI nuclear staining. Slides were mounted using ProLong™ Diamond Antifade Mountant (Invitrogen Cat#P36961, Waltham, MA, USA) and left to dry overnight. Visualization of slides was performed using the Vectra^®^ Polaris™ and analysis was conducted using Phenochart Whole Slide Viewer and Inform^®^ Tissue Analysis Software.

### 2.6. Calcein Release Assay

K562 cells were stained with Calcein-AM (Cat# 425201, BioLegend, San Diego, CA, USA) and incubated with PBMCs derived from human peripheral blood or splenocytes from Hu-NSG and Hu-NSG-Tg(IL-15) mice at varying effector-to-target ratios. The percentages of effector (NK) cells in PBMCs were determined by flow cytometry staining prior to co-culture. K562 lysis was measured 4 h after co-culture by measuring the absorbance at 528/20 nm using the SpectraMax^®^ iD3 microplate reader. Raji cell lysis was measured 6 h after co-culture with anti-CD20 antibody (BioXCell, Lebanon, NH, USA, clone 2H7, catalog BE2076) by measuring the absorbance at 528/20 nm using the SpectraMax^®^ iD3 microplate reader.

### 2.7. qPCR Viral RNA Load

Mouse peripheral blood was drawn by retro-orbital bleeding into glass capillary tubes. After a 30 min incubation at room temperature, sera was isolated by centrifugation at 1000× *g* for 15 min. Viral RNA was extracted from mouse sera using the QIAamp MinElute Virus Spin Kit (Qiagen, Hilden, Germany). Next, 1-step RT-qPCR was performed using the qScript XLT 1-Step RT-qPCR ToughMix with the primers/probe (5′-CCTGTACCGTCAGCGTTATT-3′), (5′-CAAAGAGAAGAGTGGTGGAGAG-3′) and (6-FAM-5′TG CTT CCT G-ZEN-C TGC TCC TAA GAA CC-3′-IABkFQ).

### 2.8. RNAscope

FFPE liver, spleen, and intestinal tissue of HIV-1 infected mice were used for the manual RNAscope^TM^ 2.5 HD Red assay (Cat# 322350) following the manufacturer’s protocol. The boiling method was used for target retrieval according to the recommended treatment conditions for each tissue type. The RNAscope^®^ Probe V-HIV-1-Clade A (Cat# 416101) was used to target HIV-1 RNA, RNAscope^®^ Positive Control Probe Mm-Ppib-Mus musculus peptidylprolyl isomerase B (Ppib); mRNA (Cat# 313911) was used as a positive control; and RNAscope^®^ Negative Control Probe (Cat#310043), which consists of DapB-Bacillus subtilis strain SMY methylglyoxal synthase (mgsA) gene, dihydrodipicolinate reductase (dapB) gene, and YpjD (ypjD) gene, was used as a negative control for each tissue. Quantification of RNAscope staining intensity was performed using the pixel classifier feature in QuPath [76].

### 2.9. Statistics

Statistical analysis was performed using GraphPad Prism 9 (GraphPad Software, San Diego, CA, USA). The Student’s *t*-test was used when comparing two groups, while the two-way analysis of variance (ANOVA) was used when comparing three or more groups.

### 2.10. Study Approval

All protocols involving the use of experimental animals in this study were approved by The Scripps Research Institute (TSRI) Institutional Animal Care and Use Committee and were consistent with the National Institutes of Health Guide for the Care and Use of Laboratory Animals.

## 3. Results

### 3.1. Hu-NSG-Tg(IL-15) Mice Maintain Physiological Levels of Human IL-15 and Show Efficient Long-Term Engraftment with Human Immune Cells

We determined the serum levels of human IL-15 in Hu-NSG-Tg(IL-15) mice (Figure 1A) by ELISA. We found an average of 13.4 +/− 4.9 pg/mL human IL-15 in sera of four unrelated donor cohorts (Figure 1B and Appendix A). These levels are moderately higher than average levels found in human sera (Appendix A), although human IL-15 levels are considered pathologically elevated only above 20 pg/mL [69]. We next determined the human immune cell reconstitution rate of Hu-NSG and Hu-NSG-Tg(IL-15) mice by flow cytometry analysis of their peripheral blood (PB). Reconstitution of human immune cells (mCD45^-^hCD45^+^) was robust and remained consistently high at above 80% of the total immune cell population (hCD45^+^ as a percentage of total CD45^+^ (mCD45^+^ single positive cells + hCD45^+^ single positive cells)), reaching a plateau at approximately 3 months post-transplantation with human UCB-derived CD34^+^ stem cells and remaining consistent for at least 9 months (Figure 1C and Appendix A). We next set out to identify any differences in systemic immune cell reconstitution (hCD45^+^) between Hu-NSG and Hu-NSG-Tg(IL-15) mice. Flow cytometry analysis showed no significant differences in immune cell reconstitution in the peripheral blood, spleen, liver, bone marrow, and GI tract between these two humanized mouse models. However, as previously reported for Hu-NSG mice [77], we found that Hu-NSG and Hu-NSG-Tg(IL-15) mice had minimal human immune cell reconstitution of their GI tracts (Figure 1D). We next determined the levels of circulating NK cells in NSG-Tg(IL-15) mice and compared the results with human PBMCs. We found that Hu-NSG-Tg(IL-15) mice and human PBMC had similar levels of circulating NK cells, in contrast to Hu-NSG mice, in which PB-NK cell frequencies were drastically reduced (Figure 1E). We also examined mice for any potential effects of human IL-15 expression on non-NK cell immune cell types. However, the levels of B cells, T cells, and NKT cells were similar in Hu-NSG and Hu-NSG-Tg(IL-15) mice but did not reach the levels seen in human PBMCs (Figure 1E). B cell and T cell compartments were largely similar in the spleens, livers, bone marrow (BM), and GI of Hu-NSG-Tg(IL-15) and Hu-NSG mice (Appendix A). However, NK-T cells were significantly higher in the livers and GI tract of Hu-NSG-Tg(IL-15) compared to Hu-NSG mice (Appendix A), perhaps suggesting a role for IL-15 in the differentiation or maintenance of NK-T cells in these tissues.

### 3.2. Hu-NSG-Tg(IL-15) Mice Show No Signs of Graft-Versus-Host Disease

GVHD is a major caveat with humanized immune system mice, and elevated levels of IL-15 have previously been linked to the rapid development of GVHD in previously studied IL-15 transgenic NOD/Shi-scid-IL2Rγ null (NOG) mice with elevated levels of human IL-15 [68,70]. GVHD is characterized by lymphocytic infiltration and sclerosis of the skin and other organs, which eventually results in death [78]. Encouragingly, no gross signs of GVHD were observed in the organs of Hu-NSG or Hu-NSG-Tg(IL-15) mice across multiple donors and up to 9 months post-transplant. We performed hematoxylin and eosin (H&E) staining of FFPE blocks of skin, liver, and GI tract (Figure 2A) from Hu-NSG and Hu-NSG-Tg(IL-15) mice. There were no histological signs of GVHD as determined by a pathologist.

CD4^+^ T cells are a major component of the lymphocyte infiltrate in GVHD [78]. In agreement with our histological analyses, we did not observe a significant difference in the percentage of CD4^+^ T cells between Hu-NSG and Hu-NSG-Tg(IL-15) mice in the peripheral blood, liver, spleen, or GI tract (Figure 2B). Moreover, while we did see a significant increase in CD4^+^ T cell percentage in the bone marrow of Hu-NSG-Tg(IL-15) mice, absolute counts revealed no differences in CD4^+^ T cells in the blood, liver, or GI tract. Only in the spleens of Hu-NSG-Tg(IL-15) mice did we find a modest but significant increase in the numbers of CD4^+^ T cells (Figure 2C). We further examined CD4^+^ T cell subsets by expression of their transcription factors T-bet, GATA3, RORyT, and FoxP3 (Appendix A), as they are required for the development and function of Th1, Th2, Th17, and regulatory T cells (Tregs), respectively [79,80,81,82,83,84,85,86,87,88]. We saw no significant differences in frequency of Th2, Th17, or Tregs between Hu-NSG-Tg(IL-15) and Hu-NSG mice (Appendix A). However, we did see increased levels of Th1 CD4^+^ T cells in the blood, spleens, and livers of Hu-NSG-Tg(IL-15) mice (Appendix A), perhaps indicating some IL-15-induced Th1 polarization. However, as we did not identify any gross or histological signs of GVHD, it is unlikely that these modest changes in CD4 T cell frequency and transcription factor expression have a direct effect in inducing GVHD in this Hu-NSG-Tg(IL-15) mouse model.

### 3.3. NK Cells Are Elevated across Multiple Organs in Hu-NSG-Tg(IL-15) Mice

To determine whether a physiological level of IL-15 is sufficient to significantly increase human NK cell reconstitution across multiple tissue compartments of Hu-NSG-Tg(IL-15) mice, we performed flow cytometry and immunohistochemistry (IHC) to compare the frequency of NK cells in peripheral blood, liver, spleen, bone marrow, and the GI tract between Hu-NSG and Hu-NSG-Tg(IL-15) mice. We found significant increases in the percentages of NK cells (CD56^+^CD3^-^) in the peripheral blood, spleen, liver, and BM of Hu-NSG-Tg(IL-15) mice compared to Hu-NSG mice (Figure 1E and Figure 3A). Importantly, NK cell frequencies were similar to those reported in human PBMC, spleen, and liver [66,89,90]. Furthermore, absolute numbers of NK cells were concordantly seen to be significantly higher in the peripheral blood, liver, BM, and GI tracts of Hu-NSG-Tg(IL-15) mice compared to Hu-NSG controls. A similar trend was seen in the number of splenic NK cells, although statistical significance was not reached. We confirmed results obtained by flow cytometry using IHC staining (Figure 3D) of human NK cells in spleens and livers of Hu-NSG and Hu-NSG-Tg(IL-15) mice. Quantification of IHC-stained organs showed significant increases in NKp46 expressing cells in the livers of Hu-NSG-Tg(IL-15) mice over their Hu-NSG counterparts. IHC and quantification of NKp46 expressing cells and CD3 expressing cells (Appendix A) in Hu-NSG-Tg(IL-15) mice showed the presence of NK cells and T cells in the GI tract, albeit at lower levels than in the spleen and liver. In humans, the frequency of NK cells in the GI tract is similarly much lower than that seen in the blood, spleen, liver, and lung [91]. However, absolute numbers of NK cells in the human GI tract have not been quantified. We conclude that physiological levels of IL-15 in Hu-NSG-Tg(IL-15) mice corrects NK cell developmental defects evident in Hu-NSG mice, resulting in elevated numbers of NK cells in this model.

### 3.4. NK Cells from Hu-NSG-Tg(IL-15) Mice Have a More Mature Cytotoxic CD56^dim^CD16^bright^ Phenotype

NK cells are conventionally subdivided based on varying expression of CD56 and the FcγRIII CD16. CD56^dim^CD16^bright^ NK cells are considered mature and robustly cytotoxic, while immature CD56^bright^CD16^dim^ NK cells are less cytotoxic but produce higher levels of cytokines [92]. CD56^dim^CD16^bright^ NK cells predominate the peripheral blood, while CD56^bright^CD16^dim^ NK cells constitute only 10% of PBMCs [89,90]. However, in contrast to PBMC, NK cells in tissues are enriched in CD56^bright^CD16^dim^ NK cells and comprised of both subsets [66,89,90]. Interestingly, we observed that Hu-NSG-Tg(IL-15) mice had comparable numbers of CD56^dim^CD16^bright^ NK cells in the peripheral blood, spleens, and liver, similar to human donors [66,89]. This ratio is generally not observed in Hu-NSG mice where human NK cells are predominantly CD56^bright^CD16^dim^ (Figure 4A,B). CD57 marks a terminally differentiated subset of NK cells [93]. CD57 was expressed in similar amounts in human donors and for Hu-NSG-Tg(IL-15) mice in both CD56^bright^CD16^dim^ and CD56^dim^CD56^bright^ subsets of NK cells, with the CD56^dim^CD16^bright^ subset showing higher levels of CD57 expression (Appendix A). KIR receptors and NKG2A are critical for the education of NK cells [94]. KIR expression (KIR2DL1, KIR2DS4, KIR3DL1, and KIR2DL3) was low in CD56^bright^CD16^dim^ NK cells compared to CD56^dim^CD16^bright^ in circulating NK cells from Hu-NSG-Tg(IL-15) (Appendix A). NKG2A expression was significantly higher in the CD56^dim^CD16^bright^ subset compared to human donors (Appendix A). This confirms that the CD56^dim^CD16^bright^ population of NK cells in Hu-NSG-Tg(IL-15) is indeed a more educated and mature population. NKG2C was significantly lower in circulating CD56^dim^CD16^bright^ (Appendix A) NK cells compared to human donors. This may be due to the absence of CMV exposure in Hu-NSG-Tg(IL-15) mice, which is known to result in an expansion of NKG2C^+^ NK cells [95]. Intracellular perforin and granzyme B were moderately higher in the CD56^dim^CD16^bright^ subset compared to CD56^bright^CD16^dim^ in human donors and Hu-NSG-Tg(IL-15) mice (Appendix A), with moderately higher levels seen in Hu-NSG-Tg(IL-15) mice. IFN-γ expression was low in both compartments of healthy human donors and Hu-NSG-Tg(IL-15) mice (Appendix A). Tissue residency markers CXCR6 and CD69 showed negligible differences in the peripheral blood of human donors and Hu-NSG-Tg(IL-15) mice (Appendix A). We conclude that physiological levels of IL-15 in Hu-NSG-Tg(IL-15) mice correct defects in NK cell maturation, resulting in robust numbers of CD56^dim^CD16^bright^ NK cells in multiple tissue compartments.

We also examined the expression of the T-box transcription factors T-bet and EOMES, which play key roles in NK cell development, maturation, and function [60,89,96,97,98,99]. We observed a significant increase in the T-bet^+^EOMES^-^ expressing subsets of NK cells in the blood, livers, and spleens of Hu-NSG-Tg(IL-15) mice (Figure 4C) compared to Hu-NSG controls. A concurrent decrease in the Tbet^-^EOMES^-^ population and Tbet^-^EOMES^+^ populations was also seen in the spleens and livers of Hu-NSG-Tg(IL-15) mice (Figure 4C). Altogether, our data demonstrate that phenotypically mature NK cells develop in Hu-NSG-Tg(IL-15) mice as in humans.

### 3.5. NK Cells in Hu-NSG-Tg(IL-15) Mice Have Improved Cytotoxicity over Their Hu-NSG Counterparts

Sustained elevated levels of IL-15 can impair NK cells’ effector and cytotoxic functions [100]. Therefore, we assessed the cytotoxicity of splenic NK cells from Hu-NSG-Tg(IL-15) mice compared to healthy human adult PBMC (Figure 5A,B) or NK cells from Hu-NSG mice (Figure 5C), and their responsiveness to IL-2 stimulation (Figure 5C). We did so by co-culturing NK cells with K562 target cells that are sensitive to NK cell killing due to their lack of MHC-1 expression and robust expression of stress ligands agonistic to NK cell-activating receptors [101]. We first compared the cytolysis of human PBMC-derived NK cells against that of splenic NK cells from Hu-NSG-Tg(IL-15). Hu-NSG-Tg(IL-15)-derived splenic NK cells and human PBMC-derived NK cells showed a similar dose-dependent increase in the lysis of K562 cells with an increasing effector-to-target ratio (Figure 5A). Human PBMC-derived NK cells showed slightly higher cytotoxic activity at a 10:1 E:T ratio. However, CB-HSCs and PBMC donors are genetically unrelated, which raises the possibility that the difference observed is normal donor-to-donor variation. Moreover, staining of CD107a/Lamp-1, which translocates to the cell surface membrane of NK cells upon their degranulation, tagged similar percentages of splenic Hu-NSG-Tg(IL-15) and human PBMC-derived NK cells after a 4 h co-culture with K562 cells (Figure 5B). NK cell cytotoxicity from Hu-NSG-Tg(IL-15) mice was further improved upon IL-2 stimulation (Figure 5C). Furthermore, NK cells from Hu-NSG-Tg(IL-15) mice were more effective at killing K562 cells than those isolated from Hu-NSG mice (Figure 5C), even when both were stimulated with IL-2. To assess the capacity of NK cells from Hu-NSG-Tg(IL-15) to mediate ADCC, we co-cultured splenic NK cells from Hu-NSG-Tg(IL-15) and human donors with the Raji Burkitt lymphoma cell line. In contrast to K562 cells, Raji cells express HLA class 1 molecules which interact with inhibitory KIRs on NK cells affording partial resistance [102]. However, Raji cells are susceptible to NK-cell-mediated killing upon the addition of anti-CD20 antibodies via ADCC. We found robust killing of Raji cells by splenic NK cells and from Hu-NSG-Tg(IL-15) mice which was comparable to PBMC-derived NK cells from human donors (Appendix A). Degranulation of NK cells at endpoint coincided with the specific lysis for both groups (Appendix A). We conclude from these data that NK cells in Hu-NSG-Tg(IL-15) mice are capable of robust cytotoxicity and are responsive to IL-2 stimulation not observed in Hu-NSG mice.

### 3.6. Hu-NSG-Tg(IL-15) Mice Are Susceptible to HIV-1 Infection

While human IL-15 transgenic or knock-in models have been generated, their susceptibility to HIV-1 has only been reported for one model thus far [73]. This experiment is important, as therapeutic IL-15 stimulation reactivates HIV-1 replication and reactivates cytotoxic immune cells to kill infected cells [103,104,105]. When NK cells from persons with HIV-1 on ART are treated with therapeutic levels of IL-15, their IFN-γ production and killing capacity are restored to levels similar to that of uninfected persons, as demonstrated by the killing of HIV-infected cells treated with latency-reversing drugs [106]. Further, the activation of NK cells by N-803, an IL-15 superagonist, inhibited acute HIV-1 infection of humanized mice [103]. IL-15-treatments also increased activated cytotoxic T lymphocytes (CTLs) and NK cells, reducing SHIV in the lymph nodes of non-human primates (NHP) [104]. More recently, therapeutic levels of N-803 strongly and persistently reactivated SIV in CTL-depleted NHP [105]. Therefore, we intraperitoneally infected Hu-NSG-Tg(IL-15) mice from three genetically unrelated donor cohorts with Q23.17 HIV-1 and measured viral titers in plasma by qPCR over time. Hu-NSG-Tg(IL-15) mice were highly susceptible to HIV-1 infection (Figure 6A and Appendix A) and showed a decrease in the percentage of CD4^+^ T cells starting at 4–5 weeks post-infection and remaining low thereafter (Figure 6B). We then performed RNAscope to visualize HIV-1 RNA in situ. Positive staining for HIV-1 RNA, indicated by red punctate dots, was abundantly seen throughout the spleen, with a sparser distribution in the liver and GI tract (Figure 6C,D and Appendix A). RNAscope positivity coincided with the presence of human immune cells (hCD45^+^) in the GI tract (Figure 6C and Appendix A). Quantification of RNAscope staining intensity showed statistically significant higher staining in the spleen compared to the liver and GI tract (Figure 6D). No difference in staining intensity was observed between the liver and GI tract (Figure 6D). We conclude from these data that Hu-NSG-Tg(IL-15) mice are highly susceptible to HIV-1 infection.

### 3.7. NK Cells from Hu-NSG-Tg(IL-15) Mice Are Functional and Respond to HIV-1 Infection

To determine if NK cells from Hu-NSG-Tg(IL-15) mice produce functional anti-HIV-1 responses, we performed flow cytometry to phenotype NK cells 8 weeks post-infection. We found a significantly higher proportion of IFN-γ positive NK cells in the blood and spleen of HIV-1 infected Hu-NSG-Tg(IL-15) mice compared to uninfected donor-matched controls (Figure 7A). In contrast, liver NK cells from HIV-1 infected Hu-NSG-Tg(IL-15) mice showed a higher proportion of TNF-α over uninfected controls (Figure 7B). The proportion of perforin and granzyme B-positive NK cells was high in uninfected mice, and HIV-1 infection did not significantly increase that proportion (Figure 7C,D). There was a higher percentage of Ki6- positive NK cells in the blood and liver of HIV-1 infected mice, indicating increased proliferative capacity (Figure 7E). In addition, there was a significantly higher percentage of CD107a expressing NK cells in the livers of HIV-1 infected mice (Figure 7F). We conclude that Hu-NSG-Tg(IL-15) human NK cells respond to HIV infection by cytokine release, degranulation, and proliferation.

## 4. Discussion

NK cells originate from the common lymphoid progenitor cell and produce a rapid response to both virally infected and tumorigenic cells. Human NK cell development and survival is highly dependent on the presence of human IL-15. Current humanized mouse models utilized for HIV-1 research fail to maintain physiological levels of NK cells, which creates a barrier for the study of NK-mediated effector responses against the virus. In this study, we transplanted UCB-derived HSCs into a commercially available NSG mouse expressing the human IL-15 transgene. We found that Hu-NSG-Tg(IL-15) mice maintain physiological levels of human IL-15 and engraft efficiently with human immune cells for many months. Despite this, Hu-NSG-Tg(IL-15) mice show no signs of GVHD. NK cells are elevated across multiple organs in Hu-NSG-Tg(IL-15) mice compared to Hu-NSG control mice. Moreover, Hu-NSG-Tg(IL-15) NK cells are more cytotoxic and phenotypically mature than Hu-NSG control NK cells. Importantly, Hu-NSG-Tg(IL-15) mice are susceptible to HIV-1 infection, and their NK cells respond to HIV-1 infection with cytokine production, degranulation, and proliferation. Altogether, Hu-NSG-Tg(IL-15) mice are a valuable new model for NK-cell-focused in vivo mechanistic and phenotypic studies.

Human and murine IL-15 share 73% sequence similarity [107,108], and while the binding affinity to the IL-15Rα is similar in both species [65], binding of murine IL-15 to the human IL-15βγ_c_ complex was undetectable in in vitro inhibition assays [65]. Furthermore, in human cell proliferation assays, it was demonstrated that human IL-15 signaling through human IL-15βγ_c_ is up to 1000-fold higher than murine IL-15 [65], presenting a barrier for NK cell development and proliferation in humanized mouse models that lack the physiological levels of human IL-15. Previously, multiple groups have attempted to improve the development and increase the proliferation of NK cells in humanized mouse models through a variety of interventions. Huntington et al. [56] initially showed that exogenous injections of combined human IL-15 and human IL-15Rα expanded the NK cell population more than human IL-15 administered alone. Subsequently, Chen et al. [67] showed that hydrodynamic tail-vein injections of plasmid encoding human IL-15 led to increased IL-15 expression and a subsequent elevation of NK cell numbers for up to 1 month. While these approaches were the key in identifying the necessity of IL-15 for human NK cell expansion in humanized mice, they may not accurately mimic the low sustained physiological levels of IL-15 [69]. As a result, several groups developed transgenic or knock-in mouse models with human cytokines to mimic a human physiological environment and promote the development of NK cells from human HSCs [68,71,72].

Serum levels of IL-15 in healthy humans have previously been shown to be approximately 0.8 pg/mL (IQR 0–8.68 pg/mL) [69]. Several transgenic and knock-in mouse models have been developed attempting to replicate this level of circulating human IL-15. However, in many of these mice, the serum human IL-15 levels were much higher than that reported in humans. Matsuda et al. [29] generated IL-7 and IL-15 knock-in mice on an NSG background. IL-15 levels in this model were measured to be 87.8 pg/mL +/− 9.8 pg/mL, approximately 100 times the normal level measured in healthy humans. NK cell frequencies in these double knock-in mice were exceptionally high with a proportion of 43.1 +/− 5.4% of total hCD45 in the peripheral blood and 28 +/− 5.3% in the spleen [72]. This was a likely consequence of the high IL-15 levels, as humanized IL-7 single knock-in mice did not have elevated numbers of NK cells [72]. Katano et al. [27] developed a human IL-15 transgenic mouse on a NOG genetic background. Serum IL-15 levels in this model were on average 50 pg/mL, with a range of 20–100 pg/mL [68]. When engrafted with human PBMCs, NOG-IL-15 mice showed approximately 11.3 +/− 1.5% NK cells of total human CD45^+^ cells, as well as acute GVHD and increased mortality compared to humanized NOG mice [70]. While these two models successfully produced mice with circulating human IL-15, which in turn led to a significant NK cell expansion in multiple organs, it was discovered that such high sustained levels of human IL-15 led to NK cell dysfunction. For example, peripheral blood-derived NK cells transplanted into IL-15Tg NOG mice had reduced cytotoxic activity and IFN-γ production after 4–6 weeks [68]. However, the addition of an IL-2 transgene rescued cytotoxicity in these PB-NK humanized mice, suggesting that it may not be IL-15 overstimulation alone, but a combination of factors leading to NK cell exhaustion in these models. Similarly, Elpek et al. [12] showed that a transient treatment with IL-15/IL-15Ra (a single dose of 0.5 ug IL-15 + 3 ug IL-15Ra) in C57BL/6 mice led to the initial activation and proliferation of NK cells, but that chronic stimulation (five doses of 0.5 ug IL-15 + 3 ug IL-15Ra over 2 weeks) led to the accumulation of mature NK cells that are impaired in function. In their study, the transient activation of NK cells showed high levels of Ki67, IFN-g, TNF-a, and CD107a expression in comparison to mice stimulated with high sustained levels of IL-15. Another model was developed by Herndler-Brandstetter et al. [28], which included signal-regulatory protein α (SIRPA) knock-in mice on a *Rag2^null^ IL2rg^null^* background (SRG), in addition to an IL-15 knock-in mutation (SRG-15). While baseline serum IL-15 levels were not reported, SRG-15 mice, but not SRG mice, displayed 200 pg/mL of human IL-15 upon poly I:C treatment. SRG-15 mice also have significantly higher numbers of NK cells across multiple organs. Together, these findings suggest that while high sustained levels of IL-15 lead to NK cell dysfunction, low sustained levels of IL-15 may be more suitable for NK cell homeostasis and normal functional capacity.

We observed a significant increase in NK cell frequencies across multiple organs and increase in cytotoxicity toward K562 cells in Hu-NSG-Tg(IL-15) mice compared to Hu-NSG mice, even past 6 months post-engraftment with cord-blood-derived stem cells. A recent study, by Aryee et al. [109], also confirmed these results independently for Hu-NSG-Tg(IL-15) mice up to 3 months post stem cell transplant. Thus, physiological levels of IL-15 are sufficient for support of a robust, mature NK cell population to develop in humanized mice. Importantly, we also demonstrated successful HIV-1 infection in our humanized mice. The only other human IL-15 expressing mouse model to demonstrate such is humanized SRG-15 mice [73]. These data are important, as the IL-15R superagonist N-803 robustly and persistently reactivates SIV in non-human primates and HIV-1 in BLT mice [105], albeit the physiological levels of IL-15 present in our model are far lower than those used therapeutically in this study. These data position the Hu-NSG-Tg(IL-15) mouse as a versatile model for the study of human NK cell biology, immunotherapy, and HIV-1 infection studies.

Compared to Hu-NSG mice, we found NK cells in Hu-NSG-Tg(IL-15) mice to be predominantly mature cytotoxic CD56^dim^CD16^bright^ NK cells rather than immature cytokine-producing CD56^bright^CD16^dim^ NK cells. Circulating NK cells in adult humans are predominantly (~90%) CD56^dim^CD16^bright^ [89,90]. In contrast, in adult human liver and spleen [89,90], NK cells are comprised evenly of CD56^bright^CD16^dim^and CD56^dim^CD16^bright^ NK cell subsets. In Hu-NSG-Tg(IL-15), the distribution of these subsets is similar to humans in the spleen and peripheral blood. However, in contrast to humans, Hu-NSG-Tg(IL-15) mice had higher levels of CD56^dim^CD16^bright^ cells in their livers, indicating a general shift toward the more mature cytotoxic phenotype in this model. The improved maturity in CD56^dim^CD16^bright^ NK cells was further confirmed by the expression of CD57, a marker of maturity, and the increased expression of KIRs and NKG2A, which are critical for the education of NK cells [94,110]. This increase in NK cell maturity is further reflected in the higher proportion of perforin and granzyme B expressing NK cells in uninfected Hu-NSG-Tg(IL-15) mice. Compared to Hu-NSG mice, T-bet expression, but not EOMES, was elevated across blood, spleen, and liver NK cells in Hu-NSG-Tg(IL-15). The decrease in single-positive EOMES NK cells compared to human liver NK cells [89] also provides evidence that NK cells isolated from Hu-NSG-Tg(IL-15) liver may be similar to circulating NK cells in phenotype. Future studies could investigate whether CD56^bright^ and EOMES^+^ NK cells found in humans require a different cytokine milieu for their development.

GVHD is well characterized in the BLT mouse model [78]. GVHD not only reduces cohort sizes but, more importantly, it confounds the results of long-term HIV-1 studies with the addition of a second set of underlying pathologies. A previous study by Roychowdhury et al. [111] showed that exogenous delivery of high doses (10 ug) of IL-15, but not IL-2, increases the morbidity of GVHD in a CB17-*scid* mouse humanized with human PBMCs (Hu-PBL-SCID). The median survival time of IL-15 treated Hu-PBL-SCID mice was significantly lower at 8.4 days versus 25 days for mice treated with IL-2 [111]. Therefore, due to the increased levels of IL-15 production in Hu-NSG-Tg(IL-15) mice, we evaluated long-term reconstituted (>8 months) Hu-NSG-Tg(IL-15) mice for any signs of GVHD. It is encouraging that we did not see any gross or histological evidence of GVHD in mice across three donor cohorts. Furthermore, no expansion of CD4^+^ T cells, a typical sign of GVHD, was observed in any organs. Greenblatt et al. [78] demonstrated that CD4^+^ T cells skew from a Th17 type toward a Th1 type response during GVHD. While we did not see a significant change in frequency of Th17 CD4^+^ T cells, we did see an increase in frequency of T-bet expressing Th1 CD4^+^ cells in Hu-NSG-Tg(IL-15) mice compared to their Hu-NSG counterparts. This suggests that IL-15 can typically promote GVHD by promoting the Th1 response; however, due to the lack of human HLA-restricted T cells in this model, it is unlikely to have downstream effects.

IL-15 has garnered significant attention for its therapeutic potential in HIV-1, as IL-15 superagonists can simultaneously induce latency reversal and support the survival and antiviral function of NK cells and T cells [103,104,105,106,112]. This, however, raises concerns about the ability of a transgenic IL-15 humanized mouse’s ability to sustain HIV-1 infection similar to traditional humanized mice. In a recent study, the model developed by Herndler-Brandstetter et al. [71], was successful in maintaining HIV-1 viremia, responded to ART and displayed viral rebound on treatment interruption [73]. Additionally, SRG-15 mice, when humanized with either PBMCs or HSCs, showed strong anti-HIV-1 Fc effector functions mediated by NK cells [73]. Encouragingly, we similarly saw sustained viral replication indicated by the presence of viral RNA throughout the blood, liver, spleen, and GI tract across three donor cohorts of humanized mice when infected with a CCR5-tropic subtype A HIV-1 isolate [113]. Viremia was sustained for 8 weeks even when mice were infected 6 months post-transplant with HSCs, which bodes well for long-term studies in this model. Importantly, plasma viral titers were accompanied by a decrease in peripheral blood CD4^+^ T cell percentage, which reflects the gradual loss of CD4^+^ T cells in untreated people living with HIV-1 [114]. Quantification of HIV-1 RNA in the organs of these mice showed that the spleens in particular are a major site for viral production in Hu-NSG-Tg(IL-15) mice. Together, these studies demonstrate that human IL-15 expressing mice are able to sustain HIV-1 infection and can be used for long-term studies.

NK cells are the earliest cytolytic effector cells to expand during HIV-1 infection and are essential to host resistance to HIV-1 [14,15,16,17,18,19,20,21,22,23,24,25,26,27,28,29,31]. Moreover, variation at the KIR receptors influences the effectiveness of NK cell activity against HIV-1 [14,15,16,26,29,32,33,34,35,36,37,115]. In addition, the ADCC capacity of NK cells has been correlated with improved HIV-1 control and in vaccine-induced protection against infection [41,116]. Importantly, exposed but uninfected intravenous drug users (EU IDU) had higher NK cell cytolytic activity in comparison to healthy controls and injecting drug users (IDUs) that were infected, even months prior to infection and seroconversion [20]. These NK cells from EU IDUs also produced more IFN-γ and TNF-α, both in the presence and absence of stimulation in comparison to infected IDUs and healthy controls [20]. Ravet et al. [17] demonstrated that degranulation of NK cells, measured by surface CD107a expression, from EU IDUs was both constitutively higher and occurred at a faster rate than NK cells from HIV-1+ individuals and uninfected controls. NK cells isolated from the peripheral blood of sexually exposed but uninfected individuals were enriched compared to HIV-1 infected counterparts and expressed higher levels of intracellular IFN-γ than unexposed controls [21]. Perinatally exposed but uninfected children had NK cells with higher percentage degranulation than perinatally HIV-1 infected children [22]. Unsurprisingly, Hu-NSG-Tg(IL-15) mice showed similar increases in IFN-γ and perforin expression, and increased proliferation in circulating NK cells during HIV-1 infection. Tissue-resident NK cells from Hu-NSG-Tg(IL-15) mice also showed elevated cytokine production, proliferation, and degranulation. The capacity of this humanized model to recapitulate effects seen from human donor blood-derived NK cells is encouraging, and the ability to assess tissue-specific anti-HIV-1 responses from NK cells is particularly useful to further our understanding of NK-cell-mediated control of HIV-1, as well as their role in vaccination and therapy.

NK cells are increasingly being utilized in cancer immunotherapy, alone or in combination with immune checkpoint blockade, boosting of activating signals, or retargeting via chimeric antigen receptor expression [117]. Aryee et al. [109] recently showed functional anti-tumor responses in Hu-NSG-Tg(IL-15) mice transplanted with patient-derived-xenograft melanomas. These approaches have the potential to be translated to combat HIV-1. More recent avenues of research focus on the induction of memory NK cells following vaccination, as well as targeting “regulatory” NK cells to improve the humoral immune response [118]. While there are some caveats with the Hu-NSG-Tg(IL-15) mouse model, such as a lack of MHC restriction of T cells and differences in tissue-resident subsets of NK cells, it supports the development and survival of NK cells at levels that are not seen in the Hu-NSG or BLT mouse models. Importantly, they retain their cytotoxic abilities and show similar functional responses during HIV-1 infection as adult human NK cells. We therefore conclude that the Hu-NSG-Tg(IL-15) mouse model fulfills an unmet need and provides a valuable in vivo resource for the study of NK cell responses and therapeutic applications to control, eradicate, or prevent infection with HIV-1. Other suitable infectious disease and immuno-oncology studies may similarly benefit from the availability of this model.

## Figures and Tables

**Figure 1 viruses-15-00365-f001:**
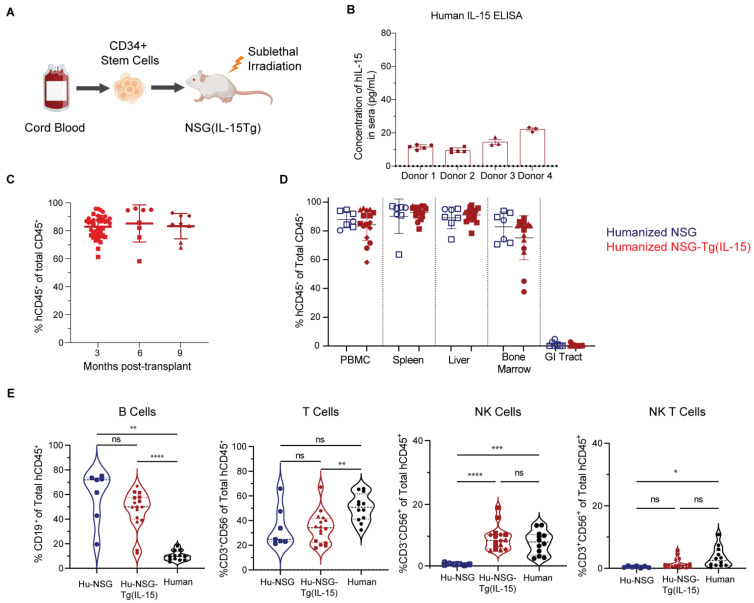
NSG-Tg(IL-15) mice support the reconstitution of human immune cells for at least 9 months post-transplant with human cord-blood-derived CD34^+^ HSCs. (**A**) Schematic of humanization of NSG-Tg(IL-15) mice with cord-blood-derived CD34^+^ stem cells. Image created with Biorender. (**B**) Concentration of human IL-15 in sera of humanized NSG-Tg(IL-15) (*n* = 16, 4 unrelated donors). (**C**) Frequency of human immune cells (hCD45^+^mCD45^-^) assessed by flow cytometry at 3, 6, and 9 months post-transplant. (*n* = 8–20, 2 unrelated donors). (**D**) Frequency of human immune cells (mCD45^-^hCD45^+^) in PBMC, spleen, liver, bone marrow, and gastrointestinal tract in humanized NSG (*n* = 7, 2 unrelated donors) and humanized NSG-Tg(IL-15) (*n* = 16, 4 unrelated donors) 6 months post-transplant with CD34^+^ stem cells. (**E**) Frequency of B cells (mCD45^-^hCD45^+^CD19^+^), T cells (mCD45^-^hCD45^+^CD3^+^CD56^-^CD19^-^), NK cells (mCD45^-^hCD45^+^CD3^-^CD56^+^CD19^-^), and NK-T cells (mCD45^-^hCD45^+^CD3^+^ CD56^+^CD19^-^) in PBMCs isolated from humanized NSG mice (*n* = 7, 2 unrelated donors), humanized NSG-Tg(IL-15) mice (*n* = 16, 4 unrelated donors) 6–9 months post CD34^+^ HSC transplant, and healthy human donors (*n* = 12). Mice were injected with 50,000 CD34+ human HSC at 6 weeks of age. Statistical significance was calculated using an unpaired t-test for 2 groups or one-way ANOVA or Brown–Forsythe test for 3 groups. Mice reconstituted from unrelated human donor cord-blood-derived CD34^+^ stem cells are denoted by different symbols. ns = not significant, * *p* < 0.05, ** *p* < 0.01, *** *p* < 0.005, and **** *p* < 0.0001.

**Figure 2 viruses-15-00365-f002:**
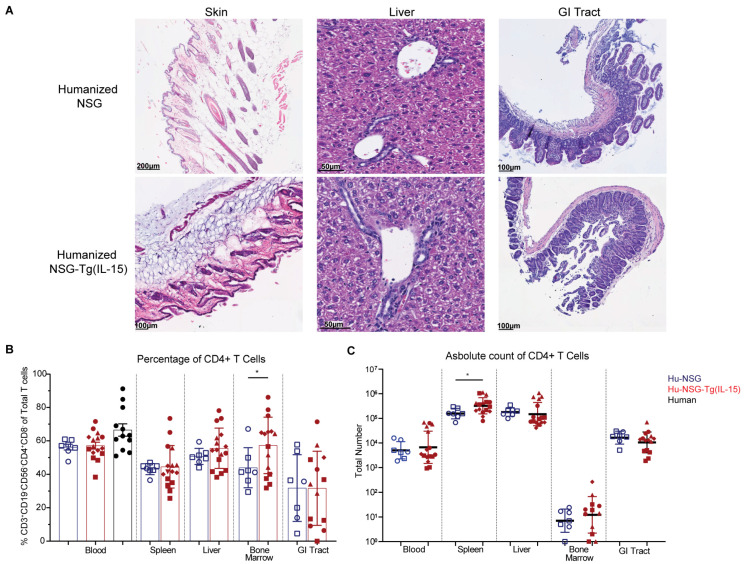
Humanized NSG-Tg(IL-15) mice do not develop signs of GVHD. (**A**) H&E staining of FFPE skin, liver, and GI from Hu-NSG and Hu-NSG-Tg(IL-15) mouse 6–9 months post-transplant with CD34^+^ stem cells. (**B**) Percentage of human CD4^+^ T cells (CD19^-^CD56^-^CD3^+^CD4^+^CD8^-^) in PBMC, spleen, liver, bone marrow, and gastrointestinal tract in Hu-NSG (*n* = 7, 2 unrelated donors) and Hu-NSG-Tg(IL-15) (*n* = 16, 4 unrelated donors). (**C**) Total number of human CD4^+^ T cells in PBMC, spleen, liver, bone marrow, and gastrointestinal tract. Mice reconstituted from unrelated human donor cord-blood-derived CD34^+^ stem cells are denoted by different symbols. Statistical significance was calculated using an unpaired *t*-test for 2 groups or Brown–Forsythe test with Dunnett’s test for multiple comparisons for 3 groups. * *p* < 0.05.

**Figure 3 viruses-15-00365-f003:**
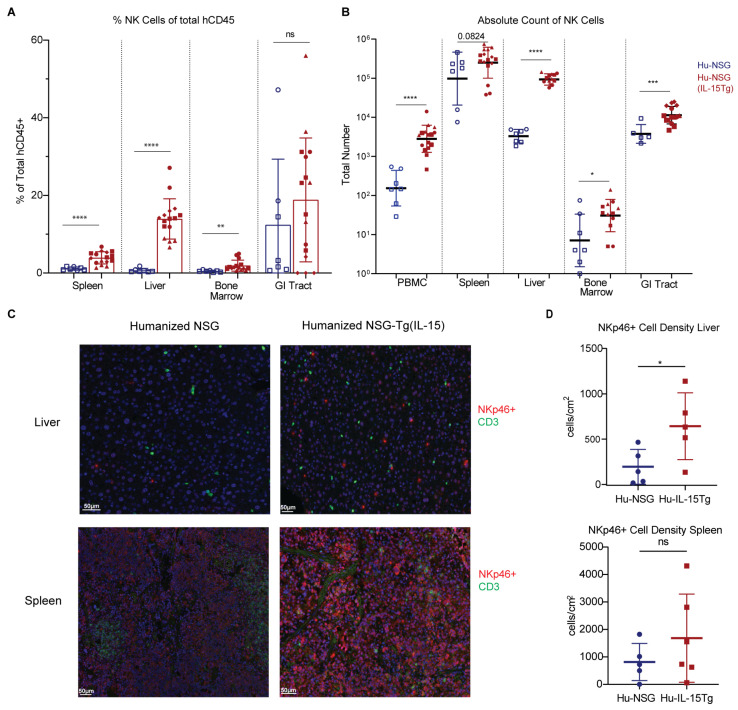
Hu-NSG-Tg(IL-15) mice support higher reconstitution of human natural killer cells than Hu-NSG mice across multiple organs. (**A**) Frequency of human NK cells in PBMC, spleen, liver, bone marrow, and gastrointestinal tract harvested from Hu-NSG (*n* = 7, 2 unrelated donors) and Hu-NSG-Tg(IL-15) (*n* = 16, 4 unrelated donors) mice 6 months post CD34^+^ HSC transplant. (**B**) Total number of human NK cells in PBMC, spleen, liver, bone marrow, and gastrointestinal tract harvested from Hu-NSG (*n* = 7, 2 unrelated donors) and Hu-NSG-Tg(IL-15) (*n* = 16, 4 unrelated donors) mice 6 months post CD34^+^ HSC transplant. Mice reconstituted from unrelated human donor cord-blood-derived CD34^+^ stem cells are denoted by different symbols. (**C**) Immunohistochemistry of FFPE NK cells (NKp46^+^) shown in red and T cells (CD3^+^) shown in green from the livers and spleens of Hu-NSG and Hu-NSG(IL-15Tg) mice 6 months post CD34^+^ HSC transplant. (**D**) Quantification of IHC-stained NKp46^+^ cells in the livers and spleens of Hu-NSG and Hu-NSG-Tg(IL-15) mice. Statistical significance was calculated using an unpaired t-test for 2 groups or Brown–Forsythe test with Dunnett’s test for multiple comparisons for 3 groups. ns = not significant,* *p* < 0.05, ** *p* < 0.01, *** *p* < 0.005, and **** *p* < 0.0001.

**Figure 4 viruses-15-00365-f004:**
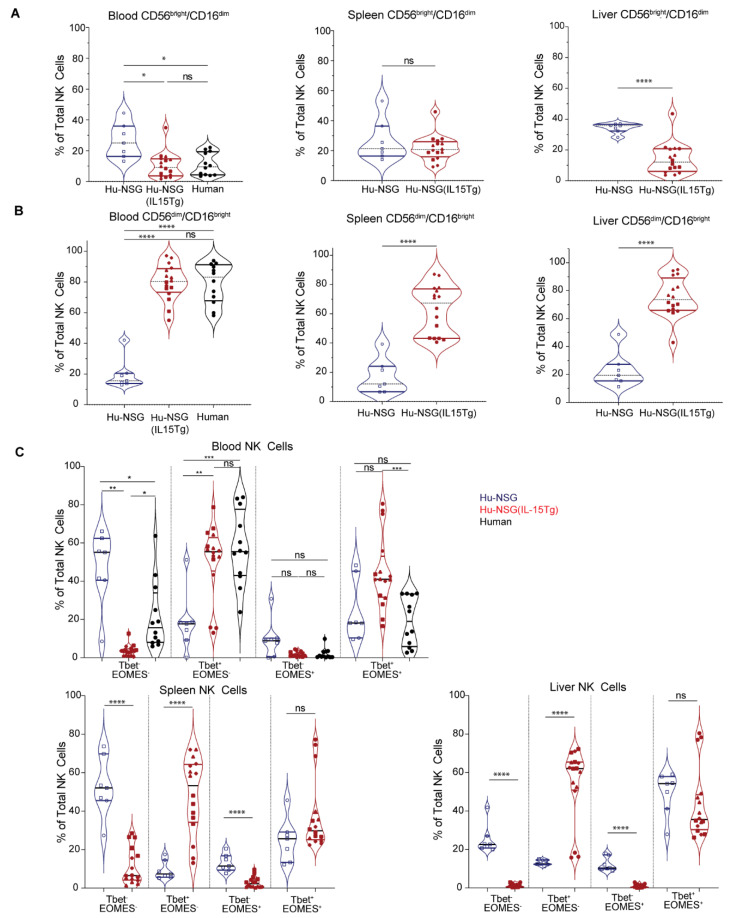
Phenotypically mature human NK cells are elevated in Hu-NSG-Tg(IL-15) mouse tissue. (**A**) Frequency of CD56^bright^CD16^dim^ of NK cells in blood, liver, and spleens of Hu-NSG mice (*n* = 7, 2 unrelated donors), Hu-NSG-Tg(IL-15) mice (*n* = 16, 4 unrelated donors) 6 months post CD34^+^ HSC transplant, and human donors (*n* = 12). (**B**) Frequency of CD56^dim^CD16^bright^ subset in blood, livers, and spleens of Hu-NSG and Hu-NSG-Tg(IL-15) mice 6 months post CD34^+^ HSC transplant. (**C**) Tbet and EOMES expression of NK cells in blood, liver, and spleens of Hu-NSG, Hu-NSG-Tg(IL-15) mice 6 months post CD34^+^ HSC transplant, and human donors. Mice reconstituted from unrelated human donor cord-blood-derived CD34^+^ stem cells are denoted by different symbols. Statistical significance was calculated using an unpaired t-test for 2 groups or Brown–Forsythe test for 3 groups. ns = not significant, * *p* < 0.05, ** *p* < 0.01, *** *p* < 0.005, and **** *p* < 0.0001.

**Figure 5 viruses-15-00365-f005:**
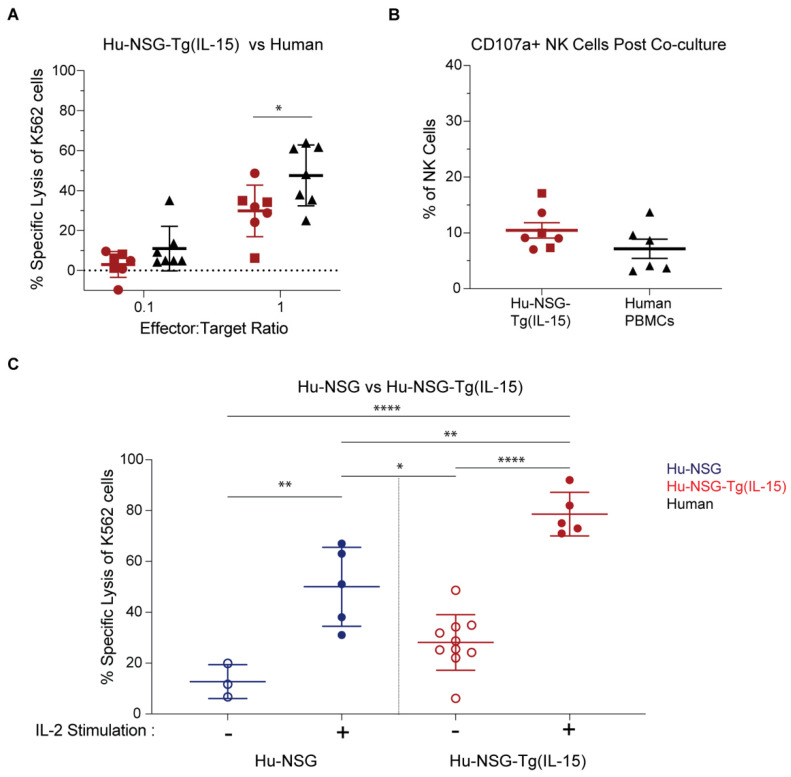
Hu-NSG-Tg(IL-15)-derived NK cells are cytotoxic. (**A**) Calcein release assay with splenocytes from Hu-NSG-Tg(IL-15) or human PBMCs co-cultured with K562 cells at 1:10 or 1:1 effector-to-target ratio. (**B**) Hu-NSG-Tg(IL-15) mouse splenocytes or human PBMCs were stained with anti-CD107a antibody after 4 h co-culture with K562 cells. CD107a^+^ expressing NK cells (hCD45^+^mCD45^-^CD3^-^CD56^+^CD107a^+^) as a percentage of total NK cells is presented. Hu-NSG-Tg(IL-15) (*n* = 7, 2 unrelated donors) and 6 human donors. Mice reconstituted from unrelated human donor cord-blood-derived CD34^+^ stem cells are denoted by different symbols. (**C**) Calcein release assay with splenocytes from Hu-NSG (*n* = 3-5, 1 donor) and Hu-NSG-Tg(IL-15) (*n* = 5-10, 1-2 donors) co-cultured with K562 cells at 1:1 effector-to-target ratio with or without IL-2 stimulation. Splenocytes from all mice were harvested mice 6 months post CD34^+^ HSC transplant. Statistical significance was calculated using an unpaired t-test for 2 groups or one-way ANOVA with multiple comparisons. * *p* < 0.05, ** *p* < 0.01, and **** *p* < 0.0001.

**Figure 6 viruses-15-00365-f006:**
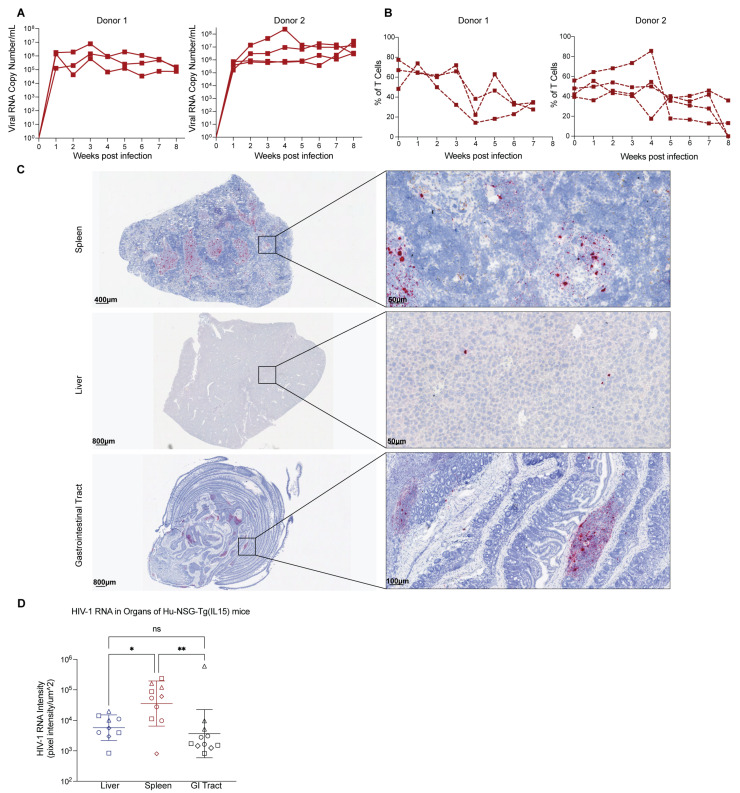
Hu-NSG-Tg(IL-15) mice are susceptible to HIV infection and demonstrate viral replication in peripheral blood, spleen, liver, and gastrointestinal tract. (**A**) RT-qPCR of viral RNA extracted from sera of HIV-1 infected (Q23.17 10^5 IU) Hu-NSG-Tg(IL-15) mice (2 donors, *n* = 3–4 per donor) 6 months post CD34^+^ stem cell transplant. (**B**) Percentage of CD4^+^ T cells of HIV-1 infected Hu-NSG-Tg(IL-15) mice (2 donors, *n* = 3–4 per donor). (**C**) Representative images from RNAscope performed on FFPE blocks of spleen, liver, and GI tract harvested 8 weeks post-infection with HIV-1 from Hu-NSG-Tg(IL-15) mice. Red punctate dots indicate positive binding of HIV-1 probe to HIV-1 RNA. (**D**) Quantification of HIV-1 RNA identified by RNAscope (3–4 donors, *n* = 2–3 per donor). Mice reconstituted from unrelated human donor cord-blood-derived CD34^+^ stem cells are denoted by different symbols. Statistical significance was calculated using an ordinary one-way ANOVA with multiple comparisons. ns = not significant, * *p* < 0.05 and ** *p* < 0.01.

**Figure 7 viruses-15-00365-f007:**
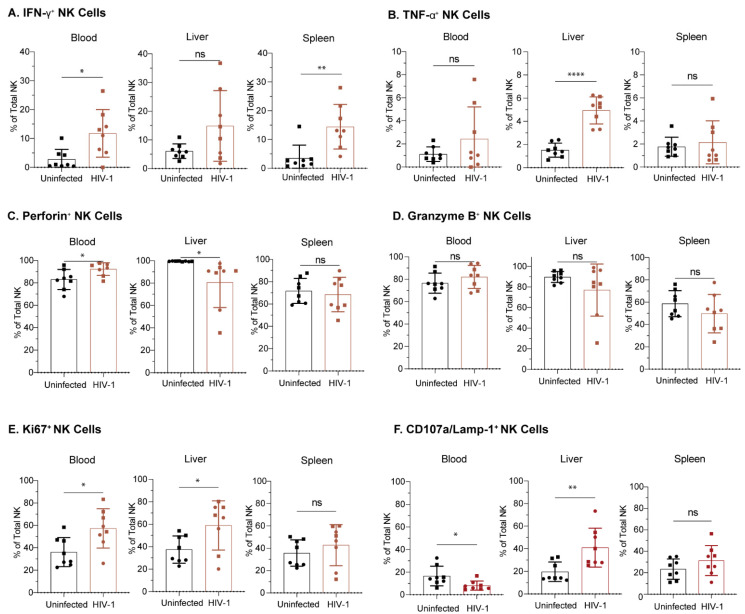
NK cells from Hu-NSG-Tg(IL-15) mice display a functional phenotype during acute HIV-1 infection. Percentage of NK cells expressing (**A**) IFN-γ, (**B**) TNF-α, (**C**) perforin, (**D**) granzyme B, (**E**) Ki67, or (**F**) CD107a (Lamp-1) in the peripheral blood, spleen, and livers of Hu-NSG-Tg(IL-15) mice, 8 weeks post HIV-1 infection. Mice from matched donors are denoted by matching symbols (*n* = 4 per donor, 2 donors per experimental group). All **Hu-NSG-Tg(IL-15)** mice were infected with HIV-1 6 months post CD34^+^ HSC transplant. Statistical significance was calculated using an unpaired t-test. ns = not significant, * *p* < 0.05, ** *p* < 0.01, and **** *p* < 0.0001.

## Data Availability

Not applicable.

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
