# Peer review of "Human Hematopoietic Stem Cell Engrafted IL-15 Transgenic NSG Mice Support Robust NK Cell Responses and Sustained HIV-1 Infection"

_viruses, 2023, doi:10.3390/v15020365_

Round 1

Reviewer 1 Report (Previous Reviewer 2)

I would like to congratulate the authors for this improved manuscript. The authors addressed most of my comments. Moreover, some of their findings have recently been corroborated/confirmed by Aryee et al., 2022.

Minor comments:

In the revised manuscript, the authors now correctly mention that the SRG-15 humanized model can also sustain HIV-1 replication. In that context, the following sentence at line 91 in the introduction must be also revised “ Also whether these IL-15 knock-in or transgenic mouse model are susceptible to HIV-1 infection has not yet been determined”

Supplemental Figure 7 legend must be corrected as this figure does not contain panel A and B.

In figure 2B, based on the figure legend, the title of Y axis should be replaced by % of CD19-CD56-CD3+CD4+CD8- cells

Author Response

Responses to Reviewer 1 Comments

I would like to congratulate the authors for this improved manuscript. The authors addressed most of my comments. Moreover, some of their findings have recently been corroborated/confirmed by Aryee et al., 2022. 

Minor comments:

In the revised manuscript, the authors now correctly mention that the SRG-15 humanized model can also sustain HIV-1 replication. In that context, the following sentence at line 91 in the introduction must be also revised “ Also whether these IL-15 knock-in or transgenic mouse model are susceptible to HIV-1 infection has not yet been determined”

We thank the reviewer for noting this oversight and have added corrections to the introduction to reflect this.

Supplemental Figure 7 legend must be corrected as this figure does not contain panel A and B.

We thank the reviewer for noting this error. We have corrected the legend by removing panel A and B and replaced it with reference to the left and right axis instead.

In figure 2B, based on the figure legend, the title of Y axis should be replaced by % of CD19-CD56-CD3+CD4+CD8- cells.

We thank the reviewer for noting this oversight. Figure 2B refers to CD19-CD56-CD3+CD4+CD8- cells as a percentage of total T cells (CD19-CD56-CD3+) and we have updated the Y axis label accordingly.

Reviewer 2 Report (New Reviewer)

The authors present a very nicely relevant paper focusing of a very very detailed description of a new humanized mouse that supports the engraftment of multilinneage human bine marrow derived immune system-but specifically NK cells.  they provide a very logical justification of why this mouse needed and demonstrate physiologic levels and phenotype of NK cells as well as other cells.  they seem very honest in pointing our subtle differences/altered physiology (such as higher levels of Th1 polarized CD4+ T cells) thus giving me confidence that its an honest report.  The model will have utility across multiple disease processes.

I have minor comments: 1)set up-the way the paper is set up it seems this will be an in depth dissection of HIV responses-yet the manuscript is an indepth description of the model.  maybe change the title a bit to something like "Human Hematopoietic Stem Cell Engrafted IL-15 Transgenic NOD.Cg-PrkdcscidIL2rgtm1wjl/Szj Mice Support Robust NK Engraftment and Cellular Responses."   Since the HIV is just one model of "validation" which could have been anything like tumors etc.  The title and intro make sit seem like this will be an HIV paper where it really a model validation paper

2)could the authors discuss one mode subtle difference/difficulty with this model-the subtleties of murine IL15 receptor components presenting and interacting with human IL15 (PMID: 12453470)    just a subtle discussion of how such xeno cytokine models may have other limitations 

Author Response

Responses to Reviewer 2 Comments

The authors present a very nicely relevant paper focusing of a very very detailed description of a new humanized mouse that supports the engraftment of multilinneage human bine marrow derived immune system-but specifically NK cells.  they provide a very logical justification of why this mouse needed and demonstrate physiologic levels and phenotype of NK cells as well as other cells.  they seem very honest in pointing our subtle differences/altered physiology (such as higher levels of Th1 polarized CD4+ T cells) thus giving me confidence that its an honest report.  The model will have utility across multiple disease processes.

I have minor comments: 1)set up-the way the paper is set up it seems this will be an in depth dissection of HIV responses-yet the manuscript is an indepth description of the model.  maybe change the title a bit to something like "Human Hematopoietic Stem Cell Engrafted IL-15 Transgenic NOD.Cg-PrkdcscidIL2rgtm1wjl/Szj Mice Support Robust NK Engraftment and Cellular Responses."   Since the HIV is just one model of "validation" which could have been anything like tumors etc.  The title and intro make sit seem like this will be an HIV paper where it really a model validation paper

We thank the reviewer for this suggestion and have edited the title to be “Humanized IL-15 Transgenic NSG Mice Support Robust Natural Killer Cell Responses and Sustained HIV-1 Infection” for improved specificity. We believe the reference to sustained HIV-1 infection is of particular importance as IL-15 can have direct (latency reversal) and indirect (NK cell mediated) effects on HIV-1 replication and is a key finding in this paper. We have also provided alternate options below if this title is not to your liking.

Other options

Human Hematopoietic Stem Cell Engrafted IL-15 Transgenic NSG Mice Support Long-Term NK cell Engraftment and Sustained HIV-1 infection.

Human Hematopoietic Stem Cell Engrafted IL-15 Transgenic NSG Mice Support Robust NK Cell Function and Sustained HIV-1 Infection.

2) Could the authors discuss one mode subtle difference/difficulty with this model-the subtleties of murine IL15 receptor components presenting and interacting with human IL15 (PMID: 12453470)    just a subtle discussion of how such xeno cytokine models may have other limitations 

We thank the reviewer for their helpful suggestion. We have added a brief description of the subtleties of the relevant cross-species IL-15 signaling in the discussion and we have included citations for PMID:12453470 in both the introduction and discussion where appropriate.

This manuscript is a resubmission of an earlier submission. The following is a list of the peer review reports and author responses from that submission.

Round 1

Reviewer 1 Report

In this manuscript, Abeynaike et al. evaluate the usage of the Hu- NSG-Tg(IL-15) mouse model to study NK cell responses to the HIV-1 infection. First, they found this mouse model expresses physiological levels of human IL-15 in serum and long-term reconstitution of human immune cells. Furthermore, they found that the reconstitution of  NK cell phenotypes is similar to humans in peripheral blood. They further showed that this mouse model does not develop graft versus host disease (GVHD) during the 9-month post-transplant of CD34 cells. Next, they showed that this model could sustain HIV-1 infection and mount human NK cell responses to HIV infection. This study reveals an important mouse model that can be used to study the role of NK cells during HIV infection in a small animal model. Thus, this study is important for the field. However, there are several concerns about experiment design and interpretation of results.

Flow cytometry studies have been done without live/dead staining. (Although authors have used SSC-A and FSC-A to discriminate viable cells from dead cells /cell debris, a significant number of dead cells will be included in the live cell population analyzed) This is one of the main concerns in the flow cytometry data in this manuscript. 

Figure 1 B:  Since two different human donors were used to generate NSG-Tg(IL15) mice. It is worth showing IL-15 levels in each group rather than adding them into one graph. (or may display each group in a different color).

Fig 2B: Better to include the human serum level of IL-15 measured under the same laboratory condition rather than compare with published literature. 

Fig 1 D. How many months after transplant? The lower level of CD45+ cells in the gut due to lower reconstitution or more cells lost/ die during sample processing? HIV is a gut tropic virus; thus, lower hCD45+ in the GIT impacts the NSG-Tg(IL15) mice model. Therefore, better to evaluate accurate gut hCD45+ cell number and cell viability.

Fig 3.C Why NKp46 used to identify NK in the IHC instead of CD56?   IL-15 can increase NKp46 expression on NK cells, misrepresent the comparison of reconstituted NK cell numbers between Hu-NSG-TG(IL15) and Hu-NSG. 

Figure 4. Comparison of NK cells in the gut and secondary lymph nodes are missing.

Figure 5. only compares the killing ability/ cytotoxicity of human NK vs. Hu-NSG(IL-15Tg) NK. However, the inclusion of NK cell cytotoxic activity of Hu-NSG would enhance the significance of this experiment. 

Figure 6. Can you present quantitative image-based analysis for the RNAscope data, which compares HIV RNA levels in 3 different tissues?  

Reviewer 2 Report

In this study, Abeynaike et al., characterizes the NSG mice expressing a transgene encoding human IL-15 (NSG-Tg(IL-15)), developed by The Jackson Laboratory. They demonstrate that these mice maintain physiological serum levels of human IL-15 and when humanized with HSCs, can support the reconstitution of human immune cells for at least 9 months post-transplant without any sign of GVHD. In contrast to most other humanized mice, these mice support the development, survival and proliferation of phenotypically mature human NK in multiple organs. Importantly, these hu-mice can sustain HIV-1 replication. While this results in NK cell activation/degranulation in vivo, the capacity of reconstituted human NK cells to mediate effectors functions against HIV-1-infected cells or to impact HIV-1 replication/pathogenesis was not clearly shown. If this the case, this hu-mice model would certainly represent a useful and accessible model to study NK cells responses against HIV-1.

Specific comments:

  • In the introduction and discussion, the authors expose the limitation of most hu-mice models that express low levels of human Il-15 that is vital for the survival and proliferation of human NK cells. They also mention that IL-15 knock-in mice have been developed but whether they support HIV-1 replication remain to be determined. Among these mice, the SRG-15 mice developed by Richard Flavell’s group was shown to support the development of functional human NK cells (PMID:29078283) and sustain HIV-1 infection (PMID:34019804). Rajashekar, Richard et al., (PMID:34019804) recently demonstrated that this hu-mice can support long-term HIV-1 infection and represent a suitable long-term model to study HIV-1 reservoir and anti-HIV-1 NK cells responses. This must be discussed by the authors. Since no NK cells exhaustion was reported with this mice model (SRG-15) upon additional stimuli (such as tumor cells engraftment or HIV-1-infection), the sentence at lines 477-478 seems inappropriate.

  • The authors clearly demonstrate that hu-NSG-Tg(IL-15) support the development of mature CD56dimCD16+ human NK cells. However, whether these NK cells are phenotypically similar to human NK cells is not demonstrated. The expression profile of key NK cell receptors (KIRs/NCRs) could be assessed and compare to human NK cells. This could certainly strengthen the conclusions of the current manuscript.

  • In the figure 5A, the cytotoxic activity of resting splenic NK cells from Hu-NSG-TG(IL-15) mice was compared to human PBMCs, but not to NK cells derived from hu-NSG mice. This comparison was only made upon IL-2 stimulation (Figure 5D). It would be important to evaluate whether the superior effector functions of NK cell derived from hu-NSG-Tg(IL-15) vs NK cell from hu-NSG mice is also observed in absence of IL-2 stimulation.

  • The authors demonstrate that hu-NSG-Tg(Il-15) mice can support HIV-1 infection (Figure 6) and conclude that this hu-mice could represent a robust model to study HIV-1 infection. To support this statement, it would be important to demonstrate that infection of hu-NSG-Tg(IL-15) mice mimic key aspect of aspect of HIV-1 pathogenesis such as CD4 T cells depletion in peripheral blood.

  • HIV-1 infection of hu-NSG-Tg(IL-15) mice resulted in NK cells activation/degranulation (Figure 7). However, the capacity of reconstituted human NK cells to directly mediate effector functions against HIV-1-infected cells or to impact HIV-1 replication/pathogenesis was not clearly shown. The capacity of reconstituted NK cells to eliminate autologous HIV-1 infected cells could be done ex vivo. The impact of NK cells on HIV-1 replication/pathogenesis in vivo could be evaluated upon NK cell depletion or by comparing viral replication in infected hu-NSG-Tg(IL-15) vs hu-NSG control mice.

  • The authors mention that this hu-mice could represents a versatile model to study immunotherapy (line 490-491). To conclude this, the capacity of NK cells derived from hu-NSG-Tg(IL-15) to mediate Fc-effector functions must be assessed.

  • The exact time post-transplant with HSCs must be added to each figure caption. This information is missing for figure 1D-E, 3, 4 and 5.